# Association between PD-L1 Expression and the Prognosis and Clinicopathologic Features of Non-Clear Cell Renal Cell Carcinoma

**DOI:** 10.3390/ijms25073916

**Published:** 2024-03-31

**Authors:** Magdalena Chrabańska, Nikola Szweda-Gandor, Magdalena Rynkiewicz, Dominik Hraboš, Bogna Drozdzowska

**Affiliations:** 1Department of Pathomorphology, Faculty of Medical Sciences in Zabrze, Medical University of Silesia, 40-055 Katowice, Poland; mrynkiewicz@sum.edu.pl (M.R.); bognadr@poczta.onet.pl (B.D.); 2Department and Clinic of Internal Medicine, Diabetology and Nephrology, Medical University of Silesia, 40-055 Zabrze, Poland; nikola.szweda@gmail.com; 3Department of Clinical and Molecular Pathology, Faculty of Medicine and Dentistry, Palacky University, 779 00 Olomouc, Czech Republic; dominik.hrabos@upol.cz

**Keywords:** PD-L1, renal cell carcinoma, papillary renal cell carcinoma, chromophobe renal cell carcinoma

## Abstract

PD-L1 is one of the two programmed cell death 1 (PD-1) ligands and a part of an immune checkpoint system (PD-1/PD-L1) with widespread clinical application. The aim of this study was to investigate PD-L1 expression and its association with clinicopathological and prognostic significance in non-clear cell renal cell carcinoma (non-ccRCC) patients. A total of 41 papillary (pRCC) and 20 chromophobe (chRCC) RCC tumors were examined for PD-L1 expression by immunohistochemistry in the cancer cells and tumor-infiltrating mononuclear cells (TIMCs). PD-L1 positivity was detected in 36.6% pRCC and 85.0% chRCC cancer cells, while PD-L1 positivity was observed in 73.2% pRCC and 50.0% chRCC TIMCs. PD-L1 positivity in both pRCC and chRCC tumor cells was not correlated with any of the examined clinicopathological features, while PD-L1 positivity in TIMCs was associated with the age of patients with pRCC. During follow-up, the death was documented among 6 patients with pRCC. Papillary RCC patients with PD-L1-positive tumor cells were significantly associated with an increased risk of death compared with patients with PD-L1-negative cancer cells. A similar trend was observed when comparing PD-L1 expression in TIMCs. However, no differences in overall survival for PD-L1-positive pRCC patients with compared to PD-L1-negative patients were observed in tumor cells or TIMCs.

## 1. Introduction

Kidney cancer is the third most common genitourinary malignancy among both sexes [1]. Renal cell carcinoma (RCC) is the most common solid lesion within the kidney, accounting for nearly 90% of all kidney malignancies. It is a heterogenous disease encompassing different histological subtypes. Clear-cell RCC (ccRCC) is the most common subtype, accounting for more than 80% of all RCCs. The remaining renal epithelial malignancies, collectively named non-clear cell RCCs (non-ccRCCs), include several subtypes such as papillary RCC (10–15%), chromophobe RCC (5%) and the more rare forms [2]. Localized tumors can be generally treated by total or partial nephrectomy. However, about 30% of patients treated with nephrectomy will still develop systemic metastases [3,4]. Some studies suggested that localized non-ccRCC cases are more likely to have a favorable prognosis than ccRCC [4,5]. In contrast, several reports claimed that some types of metastatic non-ccRCC such as papillary RCC, may have an aggressive clinical course and a shorter overall survival [6,7].

Treatment of patients with advanced RCC has progressed through the development of systemic therapy, such as immunotherapy and targeted therapy. As in other cancer types, immune checkpoint inhibitors are the focus of current research, because the immune checkpoint pathway is an important immune evasion mechanism utilized by cancer cells. PD-L1 is one of the two programmed cell death 1 (PD-1) ligands and, thus, a part of an immune checkpoint system (PD-1/PD-L1) with widespread clinical application [1,3,8].

PD-1 is a T cell co-inhibitory receptor with two ligands: PD-L1 (B7-H1) and PD-L2 (B7-DC). PD-1 is an inhibitor of both adaptive and innate immune responses, and, under normal conditions, is expressed on activated T cells, natural killer (NK) T cells, B cells, macrophages, dendritic cells and monocytes [9,10]. Of note, PD-1 is highly expressed on tumor-specific T cells. PD-1 plays two opposite roles, because it can be both beneficial and harmful. As for its beneficial effects, it reduces the regulation of ineffective or harmful immune responses and maintains immune tolerance. On the other hand, PD-1 causes the dilation of malignant cells by interfering with the protective immune response [10,11]. PD-L1 is usually expressed by macrophages, some activated T cells and B cells, dendritic cells and some epithelial cells, particularly under inflammatory conditions [12]. In addition, PD-L1 is expressed by tumor cells as an “adaptive immune mechanism” to escape anti-tumor responses [13]. Interferon gamma (IFN-γ) induces protein kinase D isoform 2 (PKD2), which is important for the regulation of PD-L1. The inhibition of PKD2 activity inhibits the expression of PD-L1 and promotes a strong antitumor immune response. T and NK cells secrete IFN-γ, increasing the expression of PD-L1 on the surface of the target cells, including tumor cells [14,15].

The PD-1/PD-L1 pathway controls the induction and maintenance of immune tolerance within the tumor microenvironment. The activity of PD-1 and its ligands PD-L1 or PD-L2 are responsible for T cell activation, proliferation and cytotoxic secretion in cancer to degenerating anti-tumor immune responses [10]. The PD-1/PD-L1 signaling axis is considered an important immunological escape mechanism exploited by neoplasms [16,17,18]. Thus, the inhibition of this pathway can enhance antitumor immune responses, leading to enhanced effector T cell activity in tissues and tumor microenvironments [8,9,17]. Based on these findings, PD-1 and its ligand PD-L1 have been investigated as novel targets in oncology [19]. Although the predictive markers for the PD-1/PD-L1 blockade remain vague, some researchers demonstrated that responsiveness and clinical outcome is better in patients with PD-L1-positive tumors detected by immunohistochemistry (IHC) [8,19,20]. Several studies reported that PD-L1 expression was associated with poor prognosis of ccRCC [8]. However, because of a relatively low occurrence and distinct biology, a significant proportion of patients with non-ccRCC have generally been excluded from the clinical trials of tumor-targeted agents. To date, only a few studies have investigated the utility of PD-L1 as a prognostic marker in non-ccRCC [3,4,8,9,21,22,23]. Thus, the prognostic value and clinical significance of PD-L1 expression in non-ccRCC subtypes still remains unclear.

Therefore, the aim of this study was to investigate PD-L1 expression and its association with clinicopathologic features and prognostic significance in non-ccRCC patients with a long-term follow-up.

## 2. Results

### 2.1. Patient Characteristics

Between 2015 and 2020, we identified 61 non-ccRCC patients with available tissue. The histological subtypes included pRCC (*n* = 41) and chRCC (*n* = 20). The clinicopathological characteristics of the patients were summarized in Table 1.

### 2.2. PD-L1 Expression in Tumor Cells and Clinicopathological Features

PD-L1 staining in tumor cells was either diffuse or focal within the tumor and showed a mosaic pattern with a random-appearing mixture of positive and negative cells. Representative images of PD-L1 expression in tumor cells were given in Figure 1 and Figure 2.

Among 61 patients with non-ccRCC, the PD-L1 expression in cancer cell membranes was negative in 29 tumors (47.5%) and positive in 32 tumors (52.5%). Specifically, PD-L1 positivity in cancer cell membranes was detected in 15 of 41 (36.6%) pRCC and 17 of 20 (85.0%) chRCC. The mean positivity of PD-L1 in non-ccRCC cells was 10.9%: 3.4% in pRCC and 27.5% in chRCC.

PD-L1 positivity in both pRCC and chRCC tumor cells was not correlated with any of the examined clinicopathological features such as age at surgery, sex, pathological tumor stage, WHO/ISUP grade, presence of tumor necrosis, angioinvasion, neuroinvasion, renal fibrous capsule invasion, perinephric fat invasion and risk of death (Table 2 and Table 3).

### 2.3. PD-L1 Expression in Tumor-Infiltrating Mononuclear Cells (TIMCs) and Clinicopathological Features

Among 61 patients with non-ccRCC, PD-L1 expression in TIMCs was negative in 21 tumors (34.4%) and positive in 40 tumors (65.6%). Specifically, PD-L1 positivity in TIMCs was detected in 30 of 41 (73.2%) pRCC and 10 of 20 (50.0%) chRCC. The mean positivity of PD-L1 in TIMCs was 7.4%: 9.6% in pRCC and 2.3% in chRCC. Representative images of PD-L1 expression in TIMCs are given in Figure 3.

Among all examined clinicopathological features, there was only a significant association of age at surgery and PD-L1 expression levels in TIMCs in pRCC (Table 2).

In the case of chRCC, PD-L1 positivity in TIMCs did not correlate with any of the examined clinicopathological features such as age during surgery, sex, pathological tumor stage, WHO/ISUP grade, presence of tumor necrosis, angioinvasion, neuroinvasion, renal fibrous capsule invasion, perinephric fat invasion and risk of death (Table 3).

### 2.4. PD-L1 Expression and Clinical Outcome in Non-ccRCC Patients

The mean duration of follow-up was 47.88 months (SD = 27.81), with a median of 43.80 months (interquartile range = 25.25 to 66.22 months).

At last follow-up, 55 (90.2%) patients were alive, while 6 (9.8%) patients died—all these cases concerned the patients with pRCC (14.6%).

Papillary RCC patients with PD-L1-positive tumor cells were significantly associated with an increased risk of death (HR = 8.20, 95% CI 1.22–6.27; *p* = 0.0012) compared with patients with PD-L1-negative tumor cells. A similar trend was observed when comparing the PD-L1 expression in TIMCs (HR = 1.33, 95% CI 1.53–12.94; *p* = 0.00032).

Kaplan–Meier analysis demonstrated no differences in overall survival (OS) for PD-L1 positive pRCC patients compared to those with PD-L1-negative tumor cells (Figure 4) and TIMCs (Figure 5).

## 3. Discussion

The main aim of cancer immunotherapies targeting the PD-1/PD-L1 pathway is to normalize the immune system instead of simply enhancing the function of immune cells in tumors. Based on the molecular mechanisms of PD-1/PD-L1 signaling, many researchers investigated various types of anti-PD-1 and anti-PD-L1 antibodies due to its clinical successful efficacy, long-lasting response and low toxicity [24,25]. Many anti-PD-1 antibodies such as nivolumab, pembrolizumab and cemiplimab have been produced to date and already approved by the FDA, while other antibodies are still in the experimental phase of development. Also several anti-PD-L1 monoclonal antibodies are commercially available. Atezolizumab, avelumab and duravulumab were approved by the FDA, but a few others are still in the experimental phase of development [24]. Since the usefulness of such immune checkpoint therapy appears limited to a specific subset of patients, there is a need to develop predictive markers to facilitate patient selection. When using anti-PD-1 or PD-L1 antibodies to treat neoplasms, some patients with low PD-L1 expression might be poor responders. Therefore, to personalize treatments and obtain an optimal treatment effect, biomarkers need to be identified. Although the FDA has approved a PD-L1 IHC test as a companion diagnostic for the immunotherapeutic targeting of the PD-1/PD-L1 pathway in some neoplasms, the expression pattern of PD-1 and PD-L1 is not a good predictive biomarker for all types of cancer [24].

Currently, there is an urgent need in particular to develop non-ccRCC-specific therapeutic approaches [26,27]. PD-L1 expression is relatively well documented in ccRCC and is reported to be associated with aggressive features such as high TNM stage, tumor size and tumor grade, as well as an increased risk of cancer-specific mortality [4,28,29,30]. The correlation between PD-L1 positivity and unfavorable prognostic factors was identified with PD-L1 expression in both tumor cells and TIMCs. Based on these studies, PD-L1 positivity may be considered an independent predictor of poor prognosis in ccRCC. However, in non-ccRCC the role of PD-L1 remains controversial.

Choueiri et al. [4] were the first to describe the PD-L1 expression in 101 non-ccRCC, including 50 pRCC and 36 chRCC, and its correlation with clinical outcome. Among these patients, PD-L1 expression in tumor cells was detected in 10.9% of cases, specifically in 10% of pRCC and 5% of chRCC. Consistent with previously published studies regarding ccRCC, they found that PD-L1 expression in non-ccRCC cells was correlated with higher tumor grade and TNM stage and shorter OS. In addition, they reported a trend of shorter OS in patients with PD-L1 positivity in TIMCs. Moreover, they observed that PD-L1 positivity in both tumor cell membranes and TIMCs was associated with a shorter time to recurrence. These observations were not confirmed in our research. We observed a much higher percentage of PD-L1 positivity in both cancer cells and TIMCs. We also did not find any associations described by Choueiri et al. [4].

Since the first research regarding non-ccRCC in 2014, only a few subsequent studies have examined PD-L1 expression in these histologic subtypes of RCC [3,8,9,21,22,23]. These studies included from 14 to 165 pRCC, from 18 to 64 chRCC and single cases of other non-ccRCC subtypes.

Chandrasekaran et al. [22] observed PD-L1 positivity in 28.6% of pRCC and 31.8% of chRCC. In their study, PD-L1 expression was significantly associated with the WHO/ISUP grading; however, it was not associated with age, gender, stage and tumor histology.

Similarly, Walter et al. [23] reported PD-L1 positivity in 32.2% of pRCC and 35.0% of chRCC. They also investigated the number of positive intratumoral cells for PD-L1 in each subtype of non-ccRCC and found that in pRCC the mean was 41.1% and in chRCC was 25.7%. In our study the mean positivity of PD-L1 was 3.4% in pRCC and 27.5% in chRCC.

Another study by Abbas et al. [9] found that the intratumoral expression of PD-L1 was observed in 46.4% non-ccRCC cases. There was no association between an intratumoral expression of PD-L1 and either patients or tumor characteristics, such as age, sex, stage, grade and lymph node or distant metastasis. Moreover, PD-L1 positivity in cancer cells was not significantly associated with cancer-specific survival (CSS) and OS in non-ccRCC; however, a trend for improved survival was seen for PD-L1-positive tumors. These observations are consistent with ours.

In the research conducted by Chipollini et al. [21], PD-L1 positivity in non-ccRCC cells was noted generally in 20% cases and in 18% of pRCC. PD-L1 positivity was significantly associated with higher nuclear grade and perineural invasion. Although not statistically significant, patients with PD-L1-positive tumors had higher AJCC pathologic tumor stages, a higher AJCC clinical M1 stage of disease, larger tumor size, more lymphovascular invasion and more tumor necrosis. Moreover, they found that PD-L1 expression was not significantly associated with 5-year CSS and 5-year relapse-free survival (RFS); however, there was a trend toward worse oncological outcomes in patients with PD-L1-positive tumors.

Shin et al. [8] investigated PD-L1 expression in a group of 201 pRCC and 10 chRCC patients and found it to be expressed in tumor cells in 6% and 10% of cases, respectively. In pRCC, PD-L1 expression did not show any significant relationship with clinicopathological variables such as age, sex, tumor size, nuclear grade, vascular invasion, necrosis, sarcomatoid transformation, lymph node metastasis and pathologic tumor stage, which corresponds with our observations.

In the latest research regarding PD-L1 expression, Möller et al. [3] investigated a large cohort of patients with RCC. In their study, PD-L1 immunostaining varied significantly between kidney cancer subtypes. This proves the well-known biological differences between different RCC subtypes. PD-L1 expression in tumor cells was 18.8% in 64 chRCC and 18.2% in 165 pRCC, which is definitely lower than in our study. In immune cells, PD-L1 expression was seen in 13.9% of pRCC and in no cases of chRCC. These results also differ significantly from ours. In pRCC, no association was found between PD-L1 expression and this cancer subtype or patient prognosis. Additionally, no association was found between PD-L1 expression in pRCC tumor cells and CD8+ density. However, there was a link between PD-L1 expression in immune cells and a high CD8+ density in pRCC.

As the above observations show, some studies concerning PD-L1 expression in cancer cells and TIMCs in non-ccRCC presented different conclusions. These partially discrepant study results are likely to have been caused by a lack of standardized procedures for PD-L1 measurement. There is no definitive cut-off value for PD-L1 positivity, with more than one threshold being reported in the literature. The variability in test cut-offs and standards for PD-L1 testing suggests that there is presently no standardized approach. According to Festino et al. [31], there is no definitive threshold result that can be universally applied to predict clinical response to PD-L1-targeted precision treatments. Published data for PD-L1 tests are mainly focused on IHC for lung cancer, while data on RCC are limited. Thereby, in our study we followed the approach verified by many researchers and based on guidelines for lung cancer where PD-L1 was considered positive when membranous tumor cell staining was observed in at least 1% of the tumor cells at any intensity [32].

Our study has several limitations. First, the non-ccRCC is a very heterogeneous disease in terms of origin and prognosis. In addition, a low incidence of the neoplasms and relatively small number of patients with these histologic subtypes have been represented, limiting our conclusions. Given the small group size for patients with number of events (deaths), a statistical analysis may not properly adjust the association of PD-L1 expression and clinical outcome for potential adverse factors. Moreover, intratumoral diversity has been described in RCC. Although we have evaluated the most representative whole tissue sections, our results may not represent the PD-L1 expression in the entire tumor. Finally, comparisons with other studies should be performed carefully, because many different methodologies and antibodies have been applied to assess PD-L1 expression.

## 4. Materials and Methods

### 4.1. Patients and Tumor Characteristics

Sixty-one patients with non-ccRCC (41 pRCC and 20 chRCC) who underwent radical or partial nephrectomy between 2015 and 2020 were included in this study. Formalin-fixed paraffin-embedded blocks were retrieved and corresponding slides from all cases were re-reviewed by two pathologists who assigned both a WHO/ISUP grade and eighth edition of the American Joint Committee on Cancer (AJCC) TNM pathological staging category. Baseline clinicopathological characteristics such as age, gender, tumor size, WHO/ISUP grade (rated only for pRCC), the presence of necrosis, sarcomatoid and rhabdoid differentiation, small vessel lymphovascular invasion, neuroinvasion, fibrous renal capsule invasion, perinephric fat invasion, renal sinus fat and vascular invasion of renal sinus vessels, macroscopic main-renal-vein invasion and AJCC TNM pathologic stage of the primary tumor (pT) were retrospectively collected for patients. The follow-up data included: date of nephrectomy, survival status, date of death and/or date of last follow-up. Anonymized and deidentified information was used for the analyses.

### 4.2. Immunohistochemical Staining

Four-micron sections were cut on glass slides and air-dried during the night. Following deparaffinization in xylene and rehydration in alcohol, heat-induced epitope retrieval was achieved by immersing the slides in buffer EnVision FLEX Target Retrieval Solution (Perlan, Beaverton, OR, USA) (pH 6.0 or 9.0) and boiling at 95 °C for 20 min. Then, slides were pre-incubated with blocking solution EnVision FLEX Peroxidase-blocking Reagent (Perlan, Beaverton, OR, USA) for 5 min. The staining was performed in an automated immunostainer according to the manufacturer’s instructions using anti-PD-L1 rabbit monoclonal antibody clone ZR3 (Zeta Corporation, Arcadia, CA, USA). The optimal dilution for antibodies was 1:100. Incubation lasted 30 min at room temperature. After this step, the sections were rinsed in buffer EnVision FLEX Wash Buffer (Perlan, Beaverton, OR, USA) and incubated with visualization system EnVision FLEX/HRP (Perlan, Beaverton, OR, USA) for 20 min. Staining patterns were visualized by exposure to 3′3-diaminobenzidine DAB (Perlan, Beaverton, OR, USA) to achieve visualization of the antigens and counterstaining with hematoxylin. Finally, the slides were dehydrated in alcohol, cleared in xylenes and mounted for examination. In each run of the experiment, replacement of the primary antibody with Tris-buffered saline was used as a negative control. The human tonsillar tissue was used as a positive control.

### 4.3. Interpretation of IHC Staining

The membranous PD-L1 staining of the cancer cells and membranous or cytoplasmic PD-L1 staining of the tumor-infiltrating mononuclear cells (TIMCs) (i.e., lymphocytes and macrophages) were separately interpreted. In normal kidney tissue, PD-L1 expression was not observed. The immunohistochemical results were independently evaluated by two pathologists blinded to the clinical outcome. PD-L1 was considered positive when IHC staining was observed in at least 1% of the cells at any intensity. Moreover, we recorded the proportion of cells stained positive in all tumor areas.

### 4.4. Statistical Analysis

The quantitative data included numbers and case percentages (%), the mean and the standard deviation (SD). The association between PD-L1 expression status and histopathological characteristics was assessed using the chi-square test and Fisher’s exact test for numerical and categorical variables, respectively. The follow-up duration was calculated from the date of surgery to the date of death or last date of follow-up. An estimator predicting the survival function was used to illustrate the Kaplan–Meier statistical analysis. A *p*-value of <0.05 was considered statistically significant. Statistical analyses were performed using Microsoft Excel 2013 (Microsoft Corporation, Redmond, WA, USA) and Statistica 13.1 (StatSoft Inc., Tulsa, OK, USA).

The standard methodology was reported in this study according to the “Strengthening the Reporting of Observational Studies in Epidemiology” (STROBE) guidelines [33].

## 5. Conclusions

In conclusion, in the era of the evolving targeted therapies and immunotargeted therapies for RCC, we have analyzed in this study PD-L1 as a novel biomarker. Our study demonstrated that in non-ccRCC, neither PD-L1-positive TIMCs nor intratumoral PD-L1 expression were associated with aggressive or advanced disease. However, the association of PD-L1 expression and risk of death was statistically significant for papillary RCC patients. This suggest that blocking the PD-1/PD-L1 pathway may be an effective method in cancer immunotherapy, and targeted therapies for PD-L1 can be of potential therapeutic benefit in the future. Therefore, further assessment within larger non-ccRCC cohorts is warranted to ascertain the relevance of PD-1/PD-L1 and prove that PD-L1 can become an independent marker for the prognosis of non-ccRCC.

## Figures and Tables

**Figure 1 ijms-25-03916-f001:**
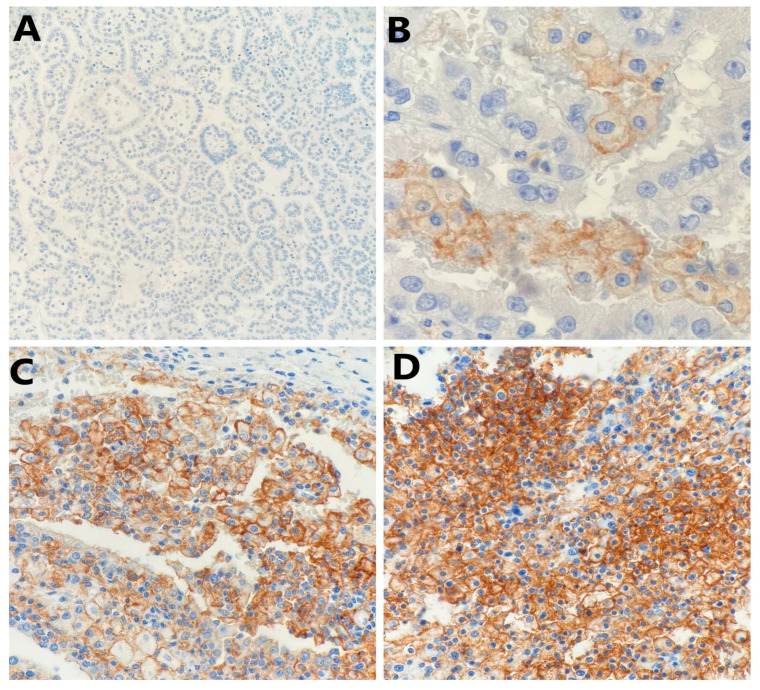
Representative photomicrography showing immunohistochemical staining of PD-L1 in papillary renal cell carcinoma tumor cells (membranous brown color): (**A**) no expression (×100), (**B**) weak expression (×400), (**C**) moderate expression (×200), and (**D**) strong expression (×200).

**Figure 2 ijms-25-03916-f002:**
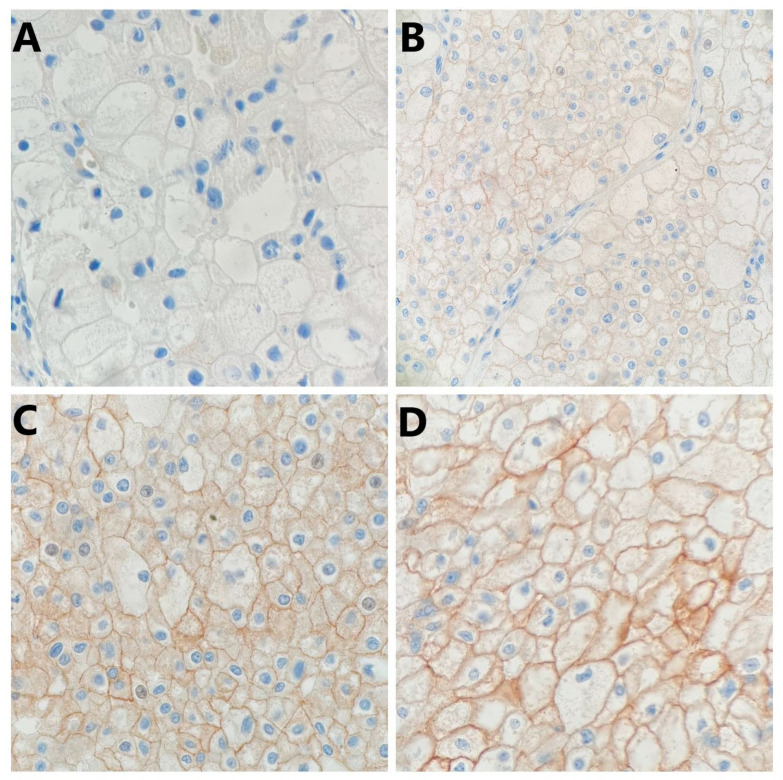
Representative photomicrography showing immunohistochemical staining of PD-L1 in chromophobe renal cell carcinoma tumor cells (membranous brown color): (**A**) no expression (×400), (**B**) weak expression (×200), (**C**) moderate expression (×400), and (**D**) strong expression (×400).

**Figure 3 ijms-25-03916-f003:**
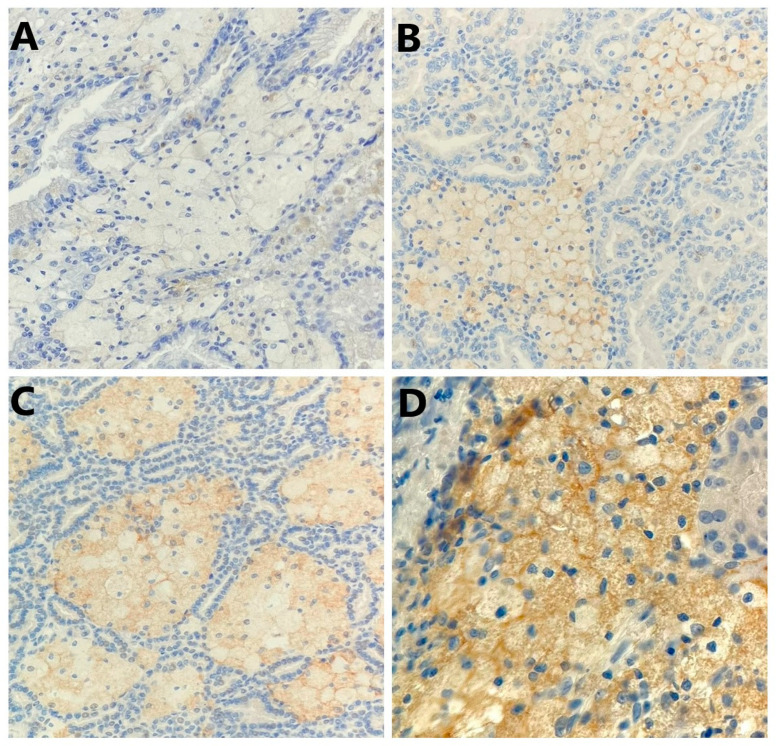
Representative photomicrography showing immunohistochemical staining of PD-L1 in the tumor-infiltrating mononuclear cells (membranous brown color): (**A**) no expression (×200), (**B**) weak expression (×200), (**C**) moderate expression (×200), and (**D**) strong expression (×400).

**Figure 4 ijms-25-03916-f004:**
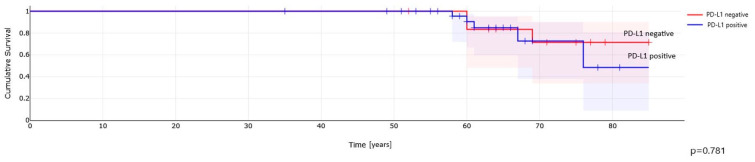
PD-L1 expression in cancer cells and overall survival in patients with papillary renal cell carcinoma.

**Figure 5 ijms-25-03916-f005:**
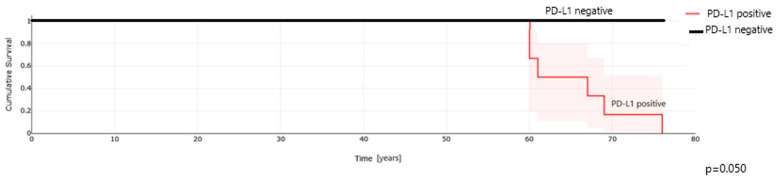
PD-L1 expression in immune cells (TIMCs) and overall survival in patients with papillary renal cell carcinoma.

**Table 1 ijms-25-03916-t001:** Non-clear cell renal cell carcinoma (RCC) patient characteristics.

Characteristics	Papillary RCC	Chromophobe RCC	Total Non-ccRCC
Number of tumor samples [*n* (%)]	41 (67.2%)	20 (32.8%)	61 (100.0%)
Age at diagnosis, years [mean ± SD]	64 ± 8	60 ± 9	62 ± 9
Gender [*n* (%)]			
Female	9 (21.9%)	7 (35.0%)	16 (26.2%)
Male	32 (78.1%)	13 (65.0%)	45 (73.8%)
Type of operation [*n* (%)]			
Radical nephrectomy	14 (34.1%)	5 (25.0%)	19 (31.1%)
Partial nephrectomy	27 (65.9%)	15 (75.0%)	42 (68.9%)
Tumor location [*n* (%)]			
Right kidney	24 (41.5%)	11 (55.0%)	35 (57.4%)
Left kidney	17 (58.5%)	9 (45.0%)	26 (42.6%)
Tumor size, cm (mean ± SD)	4.9 ± 3.3	3.8 ± 2.1	4.5 ± 1.9
Tumor stage [*n* (%)]			
pT1	26 (63.4%)	13 (65.0%)	39 (63.9%)
pT2	7 (17.1%)	2 (10.0%)	9 (14.8%)
pT3	8 (19.5%)	5 (25.0%)	13 (21.3%)
pT4	0	0	0
WHO/ISUP grading			
G1	11 (26.8%)	-	-
G2	24 (58.5%)	-	-
G3	2 (4.9%)	-	-
G4	4 (9.8%)	-	-
Tumor necrosis area % (mean ± SD)	14.5 ± 26.8	2.5 ± 7.8	10.6 ± 2.6
Sarcomatoid area % (mean ± SD)	2.5 ± 12.6	-	-
Rhabdoid area % (mean ± SD)	0	-	-
Lymphatic invasion present [*n* (%)]	4 (9.8%)	0	4 (6.3%)
Angioinvasion present [*n* (%)]	5 (12.2%)	0	5 (7.9%)
Neuroinvasion present [*n* (%)]	2 (4.9%)	0	2 (3.2%)
Renal fibrous capsule invasion present [*n* (%)]	21 (51.2%)	8 (40.0%)	29 (47.5%)
Perinephric fat invasion present [*n* (%)]	7 (17.1%)	4 (20.0%)	11 (18.0%)
Renal sinus fat invasion present [*n* (%)]	4 (9.8%)	1 (5.0%)	5 (8.2%)
Renal sinus vascular invasion present [*n* (%)]	3 (7.3%)	0	3 (4.9%)
Deaths [*n* (%)]	6 (14.6%)	0	6 (9.8%)

**Table 2 ijms-25-03916-t002:** Association of PD-L1 expression in cancer cells and immune cells (TIMCs) with the clinicopathological features in papillary renal cell carcinoma.

Feature	PD-L1 in Cancer Cells			PD-L1 in TIMCs		
	PD-L1 + *n* (%)	PD-L1 – *n* (%)	*p*-Value	PD-L1 + *n* (%)	PD-L1 – *n* (%)	*p*-Value
Age at surgery			0.74			0.02
35–45	1 (2.43%)	0		1 (2.43%)	0	
46–55	1 (2.43%)	4 (9.76%)		4 (9.76%)	1 (2.43%)	
56–65	6 (14.30%)	12 (29.27%)		15 (36.58%)	3 (7.32%)	
66–75	4 (9.76%)	7 (17.07%)		7 (17.07%)	4 (9.76%)	
76–85	3 (7.32%)	3 (7.32%)		3 (7.32%)	3 (7.32%)	
Sex			0.89			0.47
Male	10 (24.39%)	22 (53.66%)		26 (63.41%)	6 (14.30%)	
Female	5 (12.19%)	4 (9.76%)		4 (9.76%)	5 (12.19%)	
Tumor stage			0.25			0.51
pT1	9 (21.95%)	11 (26.83%)		14 (34.50%)	6 (14.30%)	
pT2	4 (9.76%)	8 (19.51%)		10 (24.39%)	2 (4.88%)	
pT3/4	2 (4.88%)	7 (17.07%)		6 (14.30%)	3 (7.32%)	
WHO/ISUP grade			0.81			0.61
G1	3 (7.32%)	8 (19.51%)		8 (19.51%)	3 (7.32%)	
G2	11 (26.83%)	13 (31.70%)		18 (43.90%)	6 (14.30%)	
G3	0	2 (4.88%)		2 (4.88%)	0	
G4	1 (2.43%)	3 (7.32%)		2 (4.88%)	2 (4.88%)	
Tumor necrosis			0.27			0.65
Present	4 (9.76%)	12 (29.27%)		10 (24.39%)	6 (14.30%)	
Absent	11 (26.83%)	14 (34.50%)		20 (48.78%)	5 (12.19%)	
Angioinvasion			0.27			0.28
Present	3 (7.32%)	2 (4.88%)		5 (12.19%)	0	
Absent	12 (29.27%)	24 (58.54%)		25 (60.97%)	11 (26.83%)	
Neuroinvasion			0.25			0.15
Present	1 (2.43%)	1 (2.43%)		2 (4.88%)	0	
Absent	14 (34.50%)	25 (60.97%)		28 (68.29%)	11 (26.83%)	
Renal fibrous capsule invasion			0.69			0.39
Present	9 (21.95%)	10 (24.39%)		14 (34.50%)	5 (12.19%)	
Absent	6 (14.30%)	16 (39.02%)		15 (36.58%)	6 (14.30%)	
Perinephric fat invasion			0.80			0.15
Present	2 (4.88%)	4 (9.76%)		6 (14.30%)	0	
Absent	13 (31.70%)	22 (53.66%)		24 (58.54%)	11 (26.83%)	
Overall survival			0.80			0.15
Death	2 (4.88%)	4 (9.76%)		6 (14.30%)	0	
Survival	13 (31.70%)	22 (53.66%)		24 (58.54%)	11 (26.83%)	

**Table 3 ijms-25-03916-t003:** Association of PD-L1 expression in cancer cells and immune cells (TIMCs) with the clinicopathological features in chromophobe renal cell carcinoma.

Feature	PD-L1 in Cancer Cells			PD-L1 in TIMCs		
	PD-L1 + *n* (%)	PD-L1 – *n* (%)	*p*-Value	PD-L1 + *n* (%)	PD-L1 – *n* (%)	*p*-Value
Age at diagnosis			0.35			0.28
35–45	1 (5%)	0		1 (5%)	0	
46–55	3 (15%)	2 (10%)		2 (10%)	3 (15%)	
56–65	7 (35%)	0		5 (25%)	2 (10%)	
66–75	6 (30%)	1 (5%)		2 (10%)	5 (25%)	
Sex			0.14			0.35
Male	12 (60%)	1 (5%)		7 (35%)	6 (30%)	
Female	5 (25%)	2 (10%)		3 (15%)	4 (20%)	
Tumor stage			0.33			0.71
pT1	9 (45%)	2 (10%)		6 (30%)	5 (25%)	
pT2	6 (30%)	1 (5%)		3 (15%)	4 (20%)	
pT3/4	2 (10%)	0		1 (5%)	1 (5%)	
Tumor necrosis			0.97			0.32
Present	3 (15%)	0		1 (5%)	2 (10%)	
Absent	15 (75%)	2 (10%)		9 (45%)	8 (40%)	
Angioinvasion			0.052			0.056
Present	0	0		0	0	
Absent	17 (85%)	3 (15%)		10 (50%)	10 (50%)	
Neuroinvasion			0.050			0.054
Present	0	0		0	0	
Absent	17 (85%)	3 (15%)		10 (50%)	10 (50%)	
Renal fibrous capsule invasion			0.81			0.39
Present	7 (35%)	1 (5%)		5 (25%)	3 (15%)	
Absent	10 (50%)	2 (10%)		5 (25%)	7 (35%)	
Perinephric fat invasion			0.73			0.13
Present	3 (15%)	1 (5%)		1 (5%)	3 (15%)	
Absent	14 (70%)	2 (10%)		9 (45%)	7 (35%)	

## Data Availability

Data are contained within the article.

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
