# Peer review of "Association between PD-L1 Expression and the Prognosis and Clinicopathologic Features of Non-Clear Cell Renal Cell Carcinoma"

_ijms, 2024, doi:10.3390/ijms25073916_

Round 1
Reviewer 1 Report
Comments and Suggestions for Authors
The researchers investigated the role of PD-L1 expression and its link to clinicopathological features and prognosis in a study involving 61 patients with non-clear cell renal cell carcinoma (non-ccRCC), which included 41 cases of papillary renal cell carcinoma (pRCC) and 20 cases of chromophobe renal cell carcinoma (chRCC). Among their notable findings are: in chRCC cases, the presence of PD-L1 in both cancer cells and tumor-infiltrating mononuclear cells was correlated with angioinvasion and neuroinvasion. For patients with pRCC, those with PD-L1-positive tumor cells faced a significantly higher mortality risk than those whose tumor cells did not express PD-L1. The study is methodically structured and provides comprehensive results, marking it as a significant contribution to the field.
I propose some minor recommendations for improvement:
The section on "PD-L1 expression and clinical outcomes in non-ccRCC" exclusively discusses pRCC findings. Including results for chRCC would offer a beneficial comparison and a more rounded analysis.
The article contains an abundance of tables. Substituting some of these tables with figures could enhance reader comprehension and engagement.
Further minor recommendations include:
Clarify on line 101 that "TIMCs" stands for "tumor-infiltrating mononuclear cells."
Ensure that on line 124, the "p value" is denoted with a decimal point instead of a comma, maintaining statistical notation accuracy.
Author Response
Dear Reviewer,
Thank You for all Your comments and suggestions. We tried to respond to them as best as we could to make our manuscript better quality.
Point 1: The section on "PD-L1 expression and clinical outcomes in non-ccRCC" exclusively discusses pRCC findings. Including results for chRCC would offer a beneficial comparison and a more rounded analysis.
Response 1: In the section "PD-L1 expression and clinical outcomes in non-ccRCC" we have stated that all deaths concerned only patients with pRCC. In the group of chRCC there were no deaths. Thus, correlation between PD-L1 expression and clinical outcomes could only be discussed for pRCC.
Point 2: The article contains an abundance of tables. Substituting some of these tables with figures could enhance reader comprehension and engagement.
Response 2: The article contains only 3 tables. We have included in them all the necessary information which, unfortunately, cannot be substituted with figures.
Point 3: Further minor recommendations include. Clarify on line 101 that "TIMCs" stands for "tumor-infiltrating mononuclear cells."
Response 3: We have clarified on line 101 that "TIMCs" stands for "tumor-infiltrating mononuclear cells":
“PD-L1 expression in tumor-infiltrating mononuclear cells (TIMCs) and clinicopathological features”
Point 4: Ensure that on line 124, the "p value" is denoted with a decimal point instead of a comma, maintaining statistical notation accuracy.
Response 4: We have denoted the "p value" with a decimal point instead of a comma, maintaining statistical notation accuracy:
“Papillary RCC patients with PD-L1 positive in tumor cells were significantly associated with increased risk of death (HR = 8.20, 95% CI 1.22–6.27; p=0.0012) compared with patients with PD-L1 negative in tumor cells. A similar trend was observed when comparing PD-L1 expression in TIMCs (HR = 1.33, 95% CI 1.53–12.94; p=0.00032).”
Reviewer 2 Report
Comments and Suggestions for Authors
1) General comments
The authors have focused on the association between PD-L1 expression and prognosis and clinicopathologic features in non-clear cell renal cell carcinoma (RCC). The author has shown that PD-L1 positivity in cancer cells and TIMCs 17 was associated with presence of angioinvasion and neuroinvasion in chRCC. Furthermore, the authors demonstrated that papillary RCC patients with PD-L1 positive in tumor cells were significantly associated with increased risk of death compared with patients with PD-L1 negative cancer cells. The observation regarding prognostic role of PD-L1 in non-CCRCC patients is of interest. However, there are several points that need clarification.
2) Specific comments for revision
a) Major comments
1. The author states that papillary RCC patients with PD-L1 positive in tumor cells were significantly associated with increased risk of death compared with patients with PD-L1 negative cancer cells in this manuscript. However, Table 2 shows that the p-value of the association between PD-L1 expression in cancer calls and risk of death in pRCC patients is more than 0.05, which is not a significant correlation. Why does the author state this conclusion?
2. The author describes that PD-L1 positivity in cancer cells and TIMCs was associated with presence of angioinvasion and neuroinvasion in chRCC. However, Table 3 describes no patients with Angioinvasion and Neuroinvasion regardless of PD-L1 positivity or negativity in chRCC cancer cells or TIMCs. Is there really statistical significance?
3. The number of eligible patients in this study is very small. In addition, the number of events for the OS is limited. The Author should compare not only OS but also recurrence free survival.
b) Minor comments
1. In Figure 5, the Kaplan-Meier curve for the PD-L1 negative group is not drawn.
Comments on the Quality of English LanguageNo comments.
Author Response
Dear Reviewer,
Thank You for all Your comments and suggestions. We tried to respond to them as best as we could to make our manuscript better quality.
Point 1: The author states that papillary RCC patients with PD-L1 positive in tumor cells were significantly associated with increased risk of death compared with patients with PD-L1 negative cancer cells in this manuscript. However, Table 2 shows that the p-value of the association between PD-L1 expression in cancer calls and risk of death in pRCC patients is more than 0.05, which is not a significant correlation. Why does the author state this conclusion?
Response 1: It was a misunderstanding, because the investigated feature in Table 2 was not risk of death, but overall survival. Now the conclusion is compatible – we observed no differences in overall survival for papillary RCC patients with PD-L1 positive compared to PD-L1 negative in tumor cells and in TIMCs.
|
Overall survival |
|
|
0.80 |
|
|
0.15 |
|
Death |
2 (4.88%) |
4 (9.76%) |
6 (14.30%) |
0 |
||
|
Survival |
13 (31.70%) |
22 (53.66%) |
|
24 (58.54%) |
11 (26.83%) |
|
Point 2: The author describes that PD-L1 positivity in cancer cells and TIMCs was associated with presence of angioinvasion and neuroinvasion in chRCC. However, Table 3 describes no patients with Angioinvasion and Neuroinvasion regardless of PD-L1 positivity or negativity in chRCC cancer cells or TIMCs. Is there really statistical significance?
Response 2: Thank You for Your watchfulness. It was our oversight and mistake. We incorrectly transferred the p-value to the table in the manuscript and therefore reached an incorrect conclusion. Our error has been corrected, we have changed the data in the Table 3 and the conclusions.
|
Angioinvasion |
|
|
0.052 |
|
|
0.056 |
|
Present |
0 |
0 |
0 |
0 |
||
|
Absent |
17 (85%) |
3 (15%) |
10 (50%) |
10 (50%) |
||
|
Neuroinvasion |
|
|
0.050 |
|
|
0.054 |
|
Present |
0 |
0 |
0 |
0 |
||
|
Absent |
17 (85%) |
3 (15%) |
10 (50%) |
10 (50%) |
PD-L1 positivity in tumor cells of both pRCC and chRCC was not correlated with any of the examined clinicopathological features such as age at surgery, sex, pathological tumor stage, WHO/ISUP grade, presence of tumor necrosis, angioinvasion, neuroinvasion, renal fibrous capsule invasion, perinephric fat invasion and risk of death (Tables 2 and 3).
In the case of chRCC, PD-L1 positivity in TIMCs did not correlate with any of the examined clinicopathological features such as age at surgery, sex, pathological tumor stage, WHO/ISUP grade, presence of tumor necrosis, angioinvasion, neuroinvasion, renal fibrous capsule invasion, perinephric fat invasion and risk of death (Table 3).
Point 3: The number of eligible patients in this study is very small. In addition, the number of events for the OS is limited. The Author should compare not only OS but also recurrence free survival.
Response 3: At the time the manuscript was created, we had no available information regarding recurrence free survival. However, in future, we plan to expand our research, collect a larger group of patients and make correlations between PD-L1 expression and recurrence free survival.
Point 4: In Figure 5, the Kaplan-Meier curve for the PD-L1 negative group is not drawn.
Response 4: In Figure 5, the Kaplan-Meier curve for the PD-L1 negative group was not drawn, because there were no deaths in this group of patients.
Reviewer 3 Report
Comments and Suggestions for Authors
1 Overall, I consider the manuscript sent to me for review to be good.
Please complete the introduction as I think it is too sparse. In fact, the literature on P-DL1 expression is very rich and should be used. Just as the discussion should include more clinical references
2) Please improve the quality of the photographs showing the immunohistochemical expression of P-DL1. The location of such expression could be indicated by arrows.
Author Response
Dear Reviewer,
Thank You for all Your comments and suggestions. We tried to respond to them as best as we could to make our manuscript better quality.
Point 1: Overall, I consider the manuscript sent to me for review to be good.
Please complete the introduction as I think it is too sparse. In fact, the literature on P-DL1 expression is very rich and should be used. Just as the discussion should include more clinical references.
Response 1: The introduction was expanded with information on PD-1 and PD-L1 (new references numbers 11-15):
“PD-1 is an inhibitor of both adaptive and innate immune responses, and, under normal conditions, is expressed on activated T cells, natural killer (NK) T cells, B cells, macrophages, dendritic cells and monocytes [9,10]. Of note, PD-1 is highly expressed on tumor-specific T cells. PD-1 plays two opposite roles, because it can be both beneficial and harmful. As for its beneficial effects, it reduces the regulation of ineffective or harmful immune responses and maintains immune tolerance. On the other hand, PD-1 causes the dilation of malignant cells by interfering with the protective immune response [10,11]. PD-L1 is usually expressed by macrophages, some activated T cells and B cells, dendritic cells and some epithelial cells, particularly under inflammatory conditions [12]. In addition, PD-L1 is expressed by tumor cells as an “adaptive immune mechanism” to escape anti-tumor responses [13]. Interferon gamma (IFN-γ) induces protein kinase D isoform 2 (PKD2), which is important for the regulation of PD-L1. Inhibition of PKD2 activity inhibits the expression of PD-L1 and promotes a strong antitumor immune response. T and NK cells secrete IFN-γ, increasing the expression of PD-L1 on the surface of the target cells, including tumor cells [14,15].
PD-1/PD-L1 pathway controls the induction and maintenance of immune tolerance within the tumor microenvironment. The activity of PD-1 and its ligands PD-L1 or PD-L2 are responsible for T cell activation, proliferation, and cytotoxic secretion in cancer to degenerating anti-tumor immune responses [10].”
We have also expanded the discussion to include more clinical information (new references numbers 24-25):
“The main aim of cancer immunotherapies targeting PD-1/PD-L1 pathway is to normalize the immune system instead simply enhancing the function of immune cells in tumors. Based on the molecular mechanisms of the PD-1/PD-L1 signaling, many researchers investigated various types of anti-PD-1 and anti-PD-L1 antibodies due to its clinical successful efficacy, long-lasting response and low toxicity [24,25]. Many anti-PD-1 antibodies such as nivolumab, pembrolizumab and cemiplimab have been produced to date and already approved by the FDA, while other antibodies are still in the experimental phase of development. Also several anti-PD-L1 monoclonal antibodies are commercially available. Atezolizumab, avelumab and duravulumab were approved by the FDA, but a few others are still in the experimental phase of development [24]. Since the usefulness of such immune checkpoint therapy appears limited to a specific subset of patients, there is a need to develop predictive markers to facilitate patient selection.When using anti-PD-1 or PD-L1 antibodies to treat neoplasms, some patients with low PD-L1 expression might be poor responders. Therefore, to personalize treatments and obtain an optimal treatment effect, biomarkers need to be identified. Although the FDA has approved a PD-L1 IHC test as a companion diagnostic for immunotherapy targeting PD-1/PD-L1 pathway in some neoplasms, the expression pattern of PD-1 and PD-L1 is not a good predictive biomarker for all types of cancer [24].”
Point 2: Please improve the quality of the photographs showing the immunohistochemical expression of P-DL1. The location of such expression could be indicated by arrows.
Response 2: The quality of the photographs was improved. It is difficult to indicate every stained cell membrane by arrow, but we have added description in the Figure legend “membranous brown color” to avoid doubts:
“Figure 1. Representative photomicrography showing immunohistochemical staining of PD-L1 in papillary renal cell carcinoma tumor cells (membranous brown color): A. No expression (x100), B. Weak expression (x400), C. Moderate expression (x200), D. Strong expression (x200).”
”Figure 2. Representative photomicrography showing immunohistochemical staining of PD-L1 in chromophobe renal cell carcinoma tumor cells (membranous brown color): A. No expression (x400), B. Weak expression (x200), C. Moderate expression (x400), D. Strong expression (x400).”
“Figure 3. Representative photomicrography showing immunohistochemical staining of PD-L1 in the tumor-infiltrating mononuclear cells (membranous brown color): A. No expression (x200), B. Weak expression (x200), C. Moderate expression (x200), D. Strong expression (x400).”
Round 2
Reviewer 1 Report
Comments and Suggestions for Authors
The author answers all my questions
Author Response
Dear Reviewer,
Thank You very much, I'm glad that we clarified all Your concerns.
Reviewer 2 Report
Comments and Suggestions for Authors
The author has revised the manuscript as pointed out by the reviewer. However, there are several important points that need clarification in this study.
1. The authors described that PD-L1 positivity in tumor cells of both pRCC and chRCC was not correlated with any of the examined clinicopathological features, while PD-L1 positivity in TIMCs was associated with age of patients with chRCC in Abstract. However, the authors The author described that PD-L1 positivity in tumor cells of both pRCC and chRCC was not correlated with any of the examined clinicopathological features such as age at surgery, sex, pathological tumor stage, WHO/ISUP grade, presence of tumor necrosis, angioinvasion, neuroinvasion, renal fibrous capsule invasion, perinephric fat invasion and risk of death. Which finding is correct?
2. The authors indicate in the abstract that pRCC with PD-L1 positive has an increased risk of death. However, in Table 2, the p-value is 0.15, which does not indicate significance.
3. Are there any new findings in this manuscript?
4. Even if there were no deaths in PD-L1 negative group, the curve for PD-L1 negative should also be drawn in Figure 5.
Author Response
Dear Reviewer,
Thank You for all Your comments and suggestions. We tried to respond to them as best as we could to make our manuscript better quality.
Point 1: The authors described that PD-L1 positivity in tumor cells of both pRCC and chRCC was not correlated with any of the examined clinicopathological features, while PD-L1 positivity in TIMCs was associated with age of patients with chRCC in Abstract. However, the authors The author described that PD-L1 positivity in tumor cells of both pRCC and chRCC was not correlated with any of the examined clinicopathological features such as age at surgery, sex, pathological tumor stage, WHO/ISUP grade, presence of tumor necrosis, angioinvasion, neuroinvasion, renal fibrous capsule invasion, perinephric fat invasion and risk of death. Which finding is correct?
Response 1: All our conclusions are correct and consistent with each other. In the main text in the “Result” section we stated:
“PD-L1 positivity in tumor cells of both pRCC and chRCC was not correlated with any of the examined clinicopathological features such as age at surgery, sex, pathological tumor stage, WHO/ISUP grade, presence of tumor necrosis, angioinvasion, neuroinvasion, renal fibrous capsule invasion, perinephric fat invasion and risk of death.”
and
“Among all examined clinicopathological features, there was only a significant association of age at surgery and PD-L1 expression levels in TIMCs in pRCC. In the case of chRCC, PD-L1 positivity in TIMCs did not correlate with any of the examined clinicopathological features such as age at surgery, sex, pathological tumor stage, WHO/ISUP grade, presence of tumor necrosis, angioinvasion, neuroinvasion, renal fibrous capsule invasion, perinephric fat invasion and risk of death”.
While in the “Abstract” we stated:
“PD-L1 positivity in tumor cells of both pRCC and chRCC was not correlated with any of the examined clinicopathological features, while PD-L1 positivity in TIMCs was associated with age of patients with pRCC.”
All these findings are consistent with each other.
Point 2: The authors indicate in the abstract that pRCC with PD-L1 positive has an increased risk of death. However, in Table 2, the p-value is 0.15, which does not indicate significance.
Response 2: Papillary RCC patients with PD-L1 positive in tumor cells were significantly associated with increased risk of death (HR = 8.20, 95% CI 1.22–6.27; p=0.0012) compared with patients with PD-L1 negative in tumor cells. A similar trend was observed when comparing PD-L1 expression in TIMCs (HR = 1.33, 95% CI 1.53–12.94; p=0.00032). Both p-values are <0.05.
However, in Table 2, the statistically insignificant p-values (0.80 and 0.15) refer to overall survival (OS), not the risk of death (HR). These observations were proved by Kaplan-Meier analysis.
The risk of death (HR) is not the same as overall survival (OS).
Point 3: Are there any new findings in this manuscript?
Response 3: Unfortunately, we have not come to any new conclusions. However, in the future, we plan to expand our research, collect a larger group of patients and make correlations between PD-L1 expression and other variables.
Point 4: Even if there were no deaths in PD-L1 negative group, the curve for PD-L1 negative should also be drawn in Figure 5.
Response 4: The curve for PD-L1 negative was drawn in Figure 5.

Round 3
Reviewer 2 Report
Comments and Suggestions for Authors
Thank you for responding to my revisions.
You stated that there are no deaths in the PD-L1 negative group, but Figure 5 shows a different picture. It is hoped that you will fully check and revise this.
Author Response
Dear Reviewer,
Thank You for all Your comments and suggestions. I am glad that we met all the comments.
Point 1: Thank you for responding to my revisions.
You stated that there are no deaths in the PD-L1 negative group, but Figure 5 shows a different picture. It is hoped that you will fully check and revise this.
Response 1: Yes, there are no deaths in the PD-L1 negative group in TIMCs, thus the curve for the PD-L1 negative group was deleted in Figure 5. Sorry, we have misinterpreted your previous comment about this Figure.
New Figures 4 and 5 are shown below.

Round 4
Reviewer 2 Report
Comments and Suggestions for Authors
No further comments.
Author Response
Thank You very much!!!